# Challenges for Sustainability in Packaging of Fresh Vegetables in Organic Farming

**Beata Michaliszyn-Gabryś [1],\*, Janusz Krupanek [1], Mariusz Kalisz [1] and Jonathan Smith [2]**

[1]  Institute for Ecology of Industrial Areas, 40-844 Katowice, Poland; j.krupanek@ietu.pl (J.K.); m.kalisz@ietu.pl (M.K.)
[2]  Scilly Organics, St Martins, Isles of Scilly TR25 0QN, UK; j.smith@scillyorganics.com
\*   Correspondence: b.michaliszyn@ietu.pl

**Abstract:** The policy of circular economy focuses on phasing out fossil-based packaging and replacing it with more sustainable alternatives. Companies face the challenge of choosing packaging for their products that are functional and affordable, and place relatively less pressure on the environment. This is especially important for organic farms that make voluntary commitments to undertake sustainable decisions regarding practices and methods of farming and types of packaging used. This publication attempts to analyze the determinants of the choices of sustainable packaging solutions made by organic farming companies with the example of Scilly Organic, an organic micro farm from the Isles of Scilly, United Kingdom—a producer of organic vegetables. There are many options for fresh vegetable packaging, which include fossil-based packaging, bio-based packaging, and packaging manufactured from material that is a mixture of synthetic, natural, or modified polymers. Biodegradable packaging, including compostable ones, is currently of particular interest because, when separated and disposed of in the correct manner in the waste management phase, they have sustainability potential. Biodegradable plastics constitute over 55.5% of global bioplastics production. Packaging is the largest market segment for bioplastic, with 48% of the total bioplastics market in 2021. Although the use of biobased packaging brings some advantages, it also comes with certain limitations that are the subject of intensive research. In this publication, the Life Cycle Assessment (LCA) tool was used and a critical review of the literature was carried out. Based on the analysis, the key factors and aspects influencing the environmental performance of selected types of packaging were identified. The LCA was carried out for the three selected packaging types, including low-density polyethylene (LDPE) bags, polylactic acid (PLA) bags, and polyester starch biopolymer (PCSB) bags. The research showed that the selection of more sustainable packaging is not straightforward. The analysis performed was the basis for providing recommendations for improving the sustainability of organic farms with regard to the selection of packaging for fresh vegetables. The critical processes in the life cycle that have to be considered are, in the first place, the production of polymer-based materials, and to a lesser extent, the production of the packaging bags and post-consumption waste utilization. In the case of PLA bags, 51% of the total impact is attributed to the production of polymer material. For starch polyester bags, this share is 58%, and for LDPE it constitutes 41% of the total score. At the same time, the choice of packaging should be made in the context of the specific properties of the packaging material, the requirements for disposal methods, and local waste management systems.

**Keywords:** sustainable packaging; bio-based packaging; organic farming; circular economy; Life Cycle Assessment (LCA)

## 1. Introduction

Packaging plays a very important role in modern economy and society, delivering a lot of benefits. On the other hand, improper design or poorly managed packaging circulation can trigger environmental problems. The principal role of food packaging is to protect food products from damage and external influences, maintain food safety, and contain food in a

cost-effective way that satisfies industry requirements and consumer desires. It also has to provide consumers with the information on the product and to minimize environmental impact [1]. Packaging is a central element to food quality preservation and extending food shelf-life [2]. Packaging was recently identified as an essential element to address the key challenge of sustainable food consumption [3,4].

Widespread use of packaging in the everyday life of consumers and in economy sectors has been observed, as well as, most of all, improper handling of the packaging used, especially plastic ones. The current circular economy policy focuses on phasing out plastic packaging and replacing it with alternatives including biobased materials and compostable ones. Therefore, a serious challenge for many companies is searching for new options. This concerns, in particular, organic farms, which are encouraged by certifiers, through standards and recommendations, to include sustainability considerations such as material source, the transportation footprint, and end-of-life options within the existing infrastructure when selecting the packaging. In order to explore benefits and limitations of biobased packaging, a lot of life cycle studies are focused on comparative analysis of environmental burdens of different packaging materials and end-of-life flows for diverted food waste. The results of the research are ambiguous and, in some cases, controversial regarding its technical, social, and environmental benefits [4,5]. The presented publication's authors' motivation to undertake research was the insufficient understanding of the conditions that influence the decision-making of organic farmers regarding sustainable packaging, taking into account political conditions, the packaging market, value chains, and environmental performance.

The aim of this publication is to assess the opportunity for improving packaging systems for fresh vegetables produced on organic farms with the use of biobased packaging through the example of Scilly Organics, an organic micro farm from the Isles of Scilly, United Kingdom. This paper discusses how local conditions of biobased packaging use in relation to other life cycle phases influence the environmental performance of packaging and which aspects of the value chain should be included when organic farmers make conscious decisions regarding sustainable packaging. Background analysis of the circular economy policy considerations and requirements of sustainable packaging is included in Section 2. The aim of this section is to show the direction of current circular economy policy in this respect at the EU and UK levels. In Section 3, background analysis of the requirements, possibilities, and options regarding packaging for fresh green vegetables is presented. This analysis aims at identifying boundary conditions for making informed choices regarding existing packaging systems in the context of the Scilly Organics micro farm. Section 4 presents an environmental analysis of the selected packaging options from a life cycle perspective. The Life Cycle Assessment (LCA) method was used to identify the weak and strong points of selected types of packaging for fresh vegetables from an environmental point of view. The main factors of environmental burdens of the packaging systems were identified, as well as the potential to improve the environmental performance of the packaging with regard to all life cycle phases, taking into account the specifics of the Scilly Island waste management system. The results were discussed by taking into account environmental analyses of the packaging included in the literature. Section 5 presents the conclusions developed on the basis of the results from Sections 2–4.

## 2. Circular Economy Policy Considerations

### 2.1. The European Union Perspective

The amount of material used for packaging is growing continuously, and in 2019 packaging waste in Europe reached a record of 177 kg per inhabitant [6]. In order to ensure that all packaging in the EU market is reusable or recyclable in an economically viable way, the Commission has undertaken measures to support the achievement of this target by 2030.

The concept of circular economy (CE) combines the well-established concept of resource efficiency while making explicit the economic aspect of saving resources and the potential gains it accrues. According to the principles of circular economy, waste is reduced

and goods are reused and recycled as much as possible. Circular economy represents the most recent attempt to conceptualize the integration of economic activity with environmental and resource concerns in a sustainable way [7]. The most widely used definition of the circular economy is the one formulated by the Ellen MacArthur Foundation in the early 2010s that states that it is an industrial system that is restorative or regenerative by intention and design. It replaces the end-of-life concept with restoration, shifts towards the use of renewable energy, eliminates the use of toxic chemicals that impair reuse, and aims for the elimination of waste through the superior design of materials, products, systems, and, within this, business models [8]. Some authors point out the need to focus not only on material preservation through recycling strategies but also to take into account the life cycle approach, including environmental, social, or economic effects connected to the materials used for packaging [9,10].

According to the New Circular Action Plan (CEAP) [11]—a continuation of the previous CEAP [12], one of the main building blocks of the European Green Deal [13]—there are several key value chains requiring urgent action, which will form an integral part of the sustainable product policy framework. Packaging and plastics are two of them. Circular economy policy plays a key role in increasing the environmental performance of packaging. The EU's transition to a circular economy is based on initiatives along the entire life cycle of products. It targets how products are designed, promotes circular economy processes, encourages sustainable consumption, and aims to ensure that waste is prevented. The document states that the Commission will review European Packaging Directive 94/62/EC to reinforce the mandatory essential requirements for packaging, with a focus on reducing (over)packaging and packaging waste, designing for the reuse and recyclability of packaging, and considering reducing the complexity of packaging materials, including the number of materials and polymers used. The work is in progress, and the expected date of completion of the document revision is July 2022.

A considerable development in the EU policy on CE was realized in 2018 when the European Commission adopted "A European Strategy for Plastics in a Circular Economy" [9,14]. Plastics are the subject of EU policy because they are a source of environmental pollution that was recognized as alarming, as polymers do not degrade but break down into smaller pieces, ending up in the air, soil, and water as microplastics [15]. The goal of this strategy is to address how plastics are designed, used, and recycled. In this document, the ambitious goal was formulated that "By 2030, all plastics packaging placed on the EU market is either reusable or can be recycled in a cost-effective manner" [9]. In this context, special attention is directed to food packaging. The EU regulation on food contact material Recital 24 [16] states that "the use of recycled materials and articles should be favoured in the community for environmental reasons, provided that strict requirements are established to ensure food safety and consumer protection". To ensure that food packaging made from recycled plastic is safe, in March 2008 the EU adopted a regulation on recycled plastic materials and articles intended to come into contact with food [17]. The regulation defines how recycled plastics can be used for food contact materials and promotes recycling and waste prevention. Apart from the above documents, the Europe's Bioeconomy Strategy, adopted in 2012, addresses the production of renewable biological resources and their conversion into vital products and bioenergy. The strategy is aimed at fossil resource replacement with sustainable natural alternatives as part of the transition to a low-carbon circular economy. As a consequence of introducing the provisions of the strategic documents, a new trend appeared on the food-packaging market. More and more companies are switching from plastic packaging to other materials. Increasingly, plastics used in the production of food packaging are being replaced by innovative biobased materials manufactured from natural resources that have potential for biodegradability, and due to their properties, could be an alternative solution [18]. The EU took measures aimed at supporting the use of biobased packaging and to improve market conditions for these products. The mandatory separate collection of biowaste will be ensured across Europe, facilitated by certified collection tools such as compostable biowaste bags. The EU policy supports the use of biobased packaging.

Biobased and recycled materials are equally viable solutions to increase the sustainability performance of packaging. According to the CEAP, the Commission will take further targeted initiatives to address the sustainability challenges posed by plastics and will continue to promote a strategy to tackle plastics pollution at a global level. The EU will support the sustainable and circular biobased sectors and strengthen them to develop substitutes to fossil-based materials that are biobased, recyclable, and marine biodegradable. With their strategy for plastics in a circular economy, the EU is forcing the industry to rethink how plastics are designed and utilized throughout the value chain to make improvements in sustainability. The European Union's regulations are transposed into the member states. Although the circular economy strategy is of a supra-national and supra-continental nature, the approach to its implementation varies by country.

The European Union circular economy policy is pushing to create more sustainable packaging and phasing out fossil-based ones in all sectors, including food production. Many initiatives have been strengthened and disseminated in the industrial context on a European scale. This is in line with organic farming strategy, strongly stimulating companies to seek new packaging solutions based on natural resources in order to meet accepted requirements and adopted goals.

### 2.2. The United Kingdom's Commitments

The United Kingdom is committed to moving towards a more circular economy, with a long history of environmental protection supported by a strong legal framework pre-dating membership in the EU. The circular economy debate in the UK has evolved over the last three to four decades from a number of converging activities, with their origins mainly in Europe [19]. The UK government has stated that leaving the EU has not changed their ambitions regarding the quality of the environment [20].

Many of the provisions covered within the European Union Circular Economy Package relate to areas of resources and waste policy where the UK nations are actively involved through existing measures to take forward commitments made in their respective domestic waste policies. The Resources and Waste Strategy (RWS) [21] for England is a part of the UK government's commitment in the 25 Year Environment Plan [22], which outlines broader steps to encourage recycling and the more thoughtful use of resources. According to the provisions of the documents, UK government policy is aimed at the reduction of plastic waste and pollution by developing a new generation of advanced and environmentally sustainable plastics, such as biobased and biodegradable packaging and bags. Regarding the packaging, the UK government is in line with the rest of the world's approach, directed towards the elimination of fossil-based materials. Nevertheless, in order to avoid piecemeal policies that encourage simple substitution of one material for another, the politics presupposes the need for a systemic approach as part of a circular economy for resources, which should ensure that material use meets three requirements: safety, sustainability, and efficiency [23]. In this context, the case of the Scilly Organics micro farm shows that making a rational decision regarding packaging options for its vegetable produce requires taking on broader perspective and life cycle thinking.

## 3. Packaging Solutions for the Organic Farm

In this section, the analysis of boundary conditions for the selection of sustainable packaging in organic farming is described, taking into account waste management considerations, with the example of the Scilly Organics farm. Organic farming is a fast-growing area in EU agriculture. It is a result of increased consumer interest in sustainable products. Organic farming is a method of producing food that aims to maintain the biological balance in the production environment by nurturing biodiversity, limiting fertilization, and increasing feed and fertilizer self-sufficiency. The basis of fertilization in plant production are organic fertilizers. The ecological production system excludes the use of synthetic substances, such as mineral fertilizers, chemical pesticides, growth hormones, and the use of genetically modified organisms and their derivatives. The EU has set out a number of

rules and regulations governing the production, distribution, and marketing of organic products in the EU. There are specific regulations related to particular products. In 2007, the EU adopted Regulation 834/2007 [24], setting out the principles, aims, and rules of organic production. The regulation is complemented by several Commission-implemented acts on the production, distribution, and marketing of organic goods. Currently, the secondary legislation is under preparation, and its entry into force was postponed by one year, from 1 January 2021 to 1 January 2022. Organic and natural foods are purchased especially by environmentally conscious consumers. Organic vegetables are premium products that can be more expensive than the alternatives. Regarding the packaging for organic food, extra design work and marketing is needed to match the characteristics of organic food processing [25].

Obtaining good functional quality of the bags depends on appropriate parameters, including the thickness, density, and weight of the material. The key functional features of the packaging are:

- Value in preserving the freshness of the vegetables: days and the need for additional measures in keeping the vegetables fresh;
- Attractiveness for consumers;
- Impact on consumer behavior regarding waste disposal;
- Opportunities and impacts on recycling options.

### 3.1. Scilly Organics Farm

Scilly Organics is a small organic micro farm situated on St. Martin's, Isles of Scilly, in United Kingdom. The Isles of Scilly is an archipelago located southwest of Cornwall. Apart from St. Martin's, the other major islands include St. Mary's, Tresco, Bryher, and St. Agnes. The farm produces fresh vegetables and fruits for local markets. The farm conducts direct sales, and the customers are local people and tourists visiting the island. The farm also sells its products to restaurants and cafes. The farm has been certified since 2004 by the Soil Association. The company is part of a circular economy demonstrator within the CIRC4Life project, focused on improving the environmental and social performance of its products and reducing waste. It is undertaking activities that engage its customers and local society in the process of circular economy practice implementation on St. Martin's.

An important challenge for Scilly Organics is sustainable waste management at the farm, as well as conscious selection of the packaging. The farm, just like other organic companies, is encouraged by the Soil Association to use the least amount of packaging possible and to use recycled or recyclable materials. The Soil Association recommends these companies take into account sustainability considerations, including end-of-life options within the existing infrastructure. Potential risks should also be considered, especially in the case of packaging waste escaping the collection systems [26].

### 3.2. Packaging Options for Scilly Organics Farm

This section presents the characteristics of the packaging recognized as suitable for fresh leafy vegetables that can be used by organic farms. The company needs packaging that consists of transparent or "milky", durable, lightweight, moisture-resistant bags and laminates to ensure safe transport for fresh produce. From the point of view of the material type meeting these criteria, the market offer includes fossil-based plastic packaging, biobased packaging, and packaging that is manufactured from material that is a mixture of synthetic and natural polymers. Various sizes of packaging for fresh vegetables are available on the market. The width of the bags for leafy vegetables is between 18 and 30 cm and the height is between 20 and 50 cm. The selection of the packaging's size depends on the content, i.e., the size of the lettuce head, the number of lettuce leaves or spinach in the package. The thickness of the bags can vary from 20 to 30 micrones.

### 3.2.1. Fossil-Based Plastic Packaging

The most popular packaging option is fossil-based plastic packaging, including polyethylene (PE) and polypropylene (PP) ones. Polyethylene includes high-density polyethylene (HDPE) and low-density polyethylene (LDPE). PP is highly crystalline and is made from thermoplastic polymer, which is a good water vapor barrier but a poor gas barrier. It is the lowest-density polymer used widely for commercial packaging, including preservation of leafy vegetables. The advantages of fossil-based plastic packaging include convenience, functionality, aesthetic appearance, availability, and affordability. PE and PP are the most common and generally used materials in food packaging because they possess excellent chemical and moisture resistance [27].

Despite the numerous benefits, fossil-based plastics are also problematic. Plastics, which take a long time to decompose and are immune to natural processes, account for a large portion of household and industrial waste (10–30%) [28]. They contain chemicals that can threaten the environment, and they need more resources to manufacture. The accumulation of plastic waste in the environment poses a serious concern. Presently, the level of awareness in society regarding the effect of plastic waste in the environment has made it necessary to reduce its impact on natural resources and decrease the emission of $CO_2$ [29]. Used fossil-based packaging must be properly disposed of by consumers. According to the European Union waste management hierarchy, they should be recycled. Where it is not possible to recycle the waste, it should be incinerated. Some fossil-based plastics, usually used in combination with starch or other bioplastics, can be engineered as biodegradable [30]. Oxo-plastics, called oxo-degradable plastics, are conventional petroleum-based polymers that include additives (to accelerate the fragmentation of the material into very small pieces, triggered by UV radiation or heat exposure). Nevertheless, LCA studies have concluded that oxo-plastic bags—in the production and use phases—do not have significantly better environmental performance than conventional polyethylene bags [31].

### 3.2.2. Biobased Packaging

Biodegradable materials manufactured from natural resources can be used as a substitute for traditional fossil-based polymers due to their easy availability and biodegradability [32]. These ecological, economic, and safety concerns have motivated researchers and industries to replace non-biodegradable polymers with biodegradable polymers and apply them as food packaging [33]. According to European Bioplastics, the term "biobased" means that the material or product is (partly) derived from biomass (plants). Biomass used for bioplastics stems from, e.g., corn, sugarcane, or cellulose. Biobased materials belong to bio-plastics—a large family of different materials. According to European Bioplastics, bioplastic refers to polymers that are either biobased, biodegradable, or feature both properties. Bioplastics or biopolymers from renewable resources are viewed by industries as a solution to environmental problems and the limited resources of petroleum-based polymers [27]. Biobased and biodegradable plastics are currently more expensive than fossil-based plastics on a weight basis. However, specific material properties can allow for cost reductions in the use or end-of-life phases [15]. Global bioplastics production capacities are set to increase from around 2.11 million tons in 2020 to approximately 2.87 million tons in 2025. The global production of biodegradable, flexible bioplastic packaging in 2020 accounted for 555,000 tons. Biodegradable plastics include PLA, PHA, starch blends, PBS, PBAT, and others, constituting over 55.5% of the global bioplastics production. The largest market segment for bioplastic is packaging with 48% (1.15 million tons) of the total bioplastics market in 2021. In bioplastics production, the major contributors are Asia (45%), Europe (25%), North America (185), and South America (12%) [34–36]. From among biobased packaging, PLA bags can be a good solution for organic farms. Specific benefits of this packaging include transparency, gloss, stiffness, printability, processability, and excellent aroma barrier. A wide range of PLA-based packaging is available on the market. Production capacity and new producers are expected to increase [37]. Polylactic acid laminations with at least one polylactic acid layer are useful for the packaging of perishable items,

including fresh produce. PLA is breathable and benefits vegetable and fruit packaging [38]. Starch blend is another option for biobased packaging for fresh produce. This material has potential environmental benefits arising either from the use of biobased feedstocks or from the desired biodegradability functionality. Starch is an affordable biomaterial and is relatively abundant, and its possibility of blending with conventional polymers has garnered wide interest in the bioplastics market [39]. Biodegradable films are characterized by poorer mechanical and water barrier properties than those made from conventional petroleum-derived polymers [40]. In order to overcome this barrier, in recent years several approaches have been investigated. Studies include, among others, reinforcing biodegradable materials with nano-materials and the multilayer strategy, in which it is possible to take advantage of the individual characteristic of the monolayer films. The currently used method is blending starch-based with conventional petroleum-derived polymers [41].

The characteristic feature of biobased packaging is its potential biodegradability, although it should be noted that fossil-based bioplastic such as polycaprolactone (PCL) is also biodegradable and, in some cases, especially when blended with starch, can be home composted. The rate of biodegradation under different biodegradation conditions is the subject of experiments and analysis [42,43]. Some biodegradable packaging could be composted in a carefully controlled environment, where factors such as source material, moisture content, temperature, oxygen levels, and acidity are monitored. Standard references such as ISO 17088:2008, EN 13432 or EN 14995, or ATM 6400 or ASTM 6868 define the specifications for compostable materials. Packaging can provide proof of compostability by meeting these standards. They are labelled by seedling label via Vinçotte or DIN CERTCO, an OK compost label via Vinçotte. Bio-degradable packaging is a good substitute for fossil-based bags, but it is also associated with shortcomings [5]. The "bio" label itself (biobased, biodegradable, bioplastic) is misunderstood by customers. Although they might interpret the "biodegradable" labeling to mean "fit for home composting", in reality, the large majority of current biodegradable plastics (e.g., PLA) can only biodegrade under very specific conditions of constantly high temperature and humidity in industrial composting installations, and they are neither fit for home composting nor do they decompose in a reasonable amount of time when littered, implying damaging consequences for fauna and flora (e.g., aquatic ones) [4]. Other doubts connected with some biobased packaging include unclear claims on environmental impacts, competition between food and non-food usage of agricultural resources, high environmental cost of some "bio" solutions, troublesome compostability of PLA, and greenwashing suspicions [4,5]. Biodegradable packaging, like any other material, must be separated from other materials for processing and disposed of in the correct manner in order to take advantage of the potential of the sustainability of this packaging. A wide range of market offer for packaging options poses a challenge to organic farmers in terms of making sustainable choices. A large variety of packaging requires clear and detailed information on compliance with relevant certificates and the composition of the material the bag is made of.

3.2.3. Market Offer of Potential Packaging for Organic Farms

In order to identify the alternative packaging for the Scilly Organics farm, the analysis of the packaging supply offer on the market was carried out, taking into account the criteria provided by Scilly Organics. According to the company, the packaging should be transparent to show the contents, compostable, and made from plant materials. Based on these criteria, several packaging bags available on the market were identified. A description of some of them can be found in Table 1, which includes information that is publicly available on the websites of the stores, including—in addition to the size and cost—information on the material from which the packaging is made, whether it is compostable, and recommendations for disposal and storage.

**Table 1.** Selected packaging bags to potentially be used by Scilly Organics as alternative solutions to fossil-based plastic bags (producers' information).

| Bag Number | Material of the Bag Packaging | Information Regarding Compostability | Storage and Disposal Instructions |
|---|---|---|---|
| Bag 1 | Wood pulp | Certified 100% commercially compostable. | Bags can be heat sealed and stored in a cool environment. They can go into consumers' food waste bins. They need to be commercially composted. |
| Bag 2 | Natural biologically sourced: potato starch and other biologically sourced polymers | Home and industrial compostable. Conforms to compostable standard EN13432. | Bags should be stored away from direct sunlight and sources of heat/humidity. Use within 12 months of delivery. Bags should only be disposed of in a controlled waste management environment. After the use phase, the bags should be placed in a food waste bin (industrial composting) or domestic compost. If composting is not available, the packaging should be placed in general waste. Not suitable for polyethene recycling. |
| Bag 3 | Natural resources: natural starches derived from thistle seed and the ear of sweetcorn. Seeds are crushed to get oil and blended with starches from corn (green ear, not the edible part). The producer does not use food sources to extract starch. | Home compostable. Conforms to OK Compost and compostable standard EN13432, Cré certification. | Store away from humidity, heat sources, and direct sunlight; use within 12 months of delivery. To be placed in an organic waste bin or home composter. |
| Bag 4 | Renewable resources | Home compostable. Certified by the compostability standards EN 13432, ASTM6400, OK Compost Home (Europe), and AS5810 (Australia). | The home compost bin is an ideal environment for decomposition. The bags should be disposed of in an environment containing heat, water, oxygen, soil, and microorganisms. |
| Bag 5 | Proprietary blends of fully compostable polymers that are both biobased (20–80%) and fossil-based (the remaining percentage) | Home compostable. Certified by the compostability standards EN 13432, ASTM D6400, AS 4736/AS 5810, and TÜV OK Compost Home. The home compostable products are certified by TÜV Austria and ABA. | Requires proper storage conditions below 30 °C and a humidity of 50%. In places with extreme humidity or heat the degradation process can be accelerated. Composting conditions: Standard home compost temperatures tend to hover at $25 \pm 5$ °C, with a target humidity of around 50%. Under normal composting conditions, the packaging will disintegrate within 6 months and fully degrade within a year. This packaging is not meant to be littered or disposed of in marine environments or land ecosystems. It should be disposed of in the proper waste stream, where it will biodegrade into compost. It is not recyclable and needs to be removed from the recycling stream. |

Of the analyzed packaging, one of them requires industrial composting (bag 1), one bag can be composted both at home and at an industrial composting plant (bag 2), and three can be composted at home (bags 3, 4, and 5). The producers of all the bags studied provide recommendations on their websites for storing the bags to ensure good quality as well as tips for handling the used packaging that should be followed by consumers who purchase fresh vegetables packaged in the bags. In the case of bags 1 and 2, which are suitable for industrial composting, after the consumption phase they should be placed in the food waste bin. Most of the bags analyzed, with the exception of bag 1, which was labelled as suitable for home composting, can be placed in a home composter. If the consumer

does not have a home composter and there is no industrial composting facility nearby, the used bag should be placed in a mixed waste container. They should not be placed in a standard recycling bin in order to not contaminate the recycling process. Compostable and biodegradable plastics currently have a low potential of recycling in practice. It should be emphasized that compostable packaging cannot be disposed of in the environment after the use phase and must be properly treated according to the manufacturer's recommendations. The research has shown that for the packaging bags studied, there is no information on the environmental impact of the packaging throughout its life cycle. In addition, the manufacturers of the packaging do not provide detailed information on the composition of the packaging material. In the case of bag 3, the manufacturer provides some details on the composition of the material upon request. The conducted analysis of the packaging supply on the market was used by Scilly Organics as an input for the selection of packaging that meets their expectations.

Biobased compostable packaging for fresh vegetables is already available on the market. They are sold in different sizes. It is also possible to adjust the dimensions of the bags to the needs of the consumer. The bags can be perforated, micro-perforated to give air to the vegetables, or non-perforated. They can be sealed or unsealed and can be sealed by the vegetable producer or supplier with a sealing device. The supply of these bags is much more limited than, for example, that of biobased packaging for take-away food. This is confirmed by literature studies, according to which, despite the extremely dynamic development and research in the field of biobased and/or biodegradable materials, commercially available bio-packaging does not yet properly meet the huge market and consumer demands [4]. The choice of an appropriate packaging solution depends on the product's specifications and the full understanding of its properties and end-of-life conditions, especially by the end consumer.

### 3.3. Postconsumption Utilization of Packaging Waste on the Isles of Scilly

In the considered area, waste management is implemented according to the Isles of Scilly Waste Reduction Strategy 2020–2030 [44]. The ambition of the strategy is to use resources efficiently, effectively, and sustainably in order to reduce the amount of waste generated and increase reuse and recycling of the material. The Council of the Isles of Scilly manages waste and recycling services for householders and businesses across four islands: St. Mary's, St. Agnes, Bryher, and St. Martin's.

Household waste collection on the Isles of Scilly is separated into two streams. Recyclable clean and dry fractions such as paper, metal, food tins and cans, plastic bottles, tubs, pots, and trays are stored together in a "recycling sack". The remaining waste (other dirty materials that cannot be recycled, foil-lined packets, diapers, any food wasted that cannot be composted at home) goes to a "waste sack" as residual waste [45]. Waste generated by households and businesses is collected by an island waste contractor on a weekly basis, weighed, and containerized. Prior to further transport, the wastes are delivered to a local waste management site at Porthmellon (St. Mary's Island), which has been redeveloped to provide a Household Waste Recycling Centre (HWRC) and transfer station. Apart from selectively collected green waste, other waste is transported and treated on the mainland; thus, managing and moving waste from the islands to the mainland, or indeed even within the islands, is expensive and presents practical challenges. Around 10 standard 20 ft shipping containers of waste are sent to the mainland on a weekly basis, which includes waste from both collected waste streams [46]. On the mainland, dry recyclables are treated under recycling processes. Residual waste is disposed of at the WTE (waste to energy) plant at St. Dennis in Cornwall [47]. Green waste is diverted to a local composting facility on St. Mary's Island, which is an open "windrows" arrangement used to generate compost for local farms or intended to be used in other applications (for example, by municipal services). A common, insular practice is home composting of organic waste and the use of compost in gardening. Postconsumer packaging waste is usually considered for recycling; however, in practice the effectiveness and profitability depend on the type of waste, the

implemented collection system, and the environmental awareness of the inhabitants, which determines the acquisition of clean raw material for further processing. Packaging that is in direct contact with food and potentially contaminated with it is difficult to recycle and can limit its effectiveness, likewise, when food waste along with fossil-based packaging goes to waste destined for composting. This "waste mix" reduces the quality of the compost and is a source of other waste that is difficult to manage (e.g., out-of-class compost contaminated with plastic). These aspects hinder the implementation of the principles of a sustainable circular economy. An alternative to this is the use of compostable packaging; thus, an LCA comparison of both fossil-based plastics and biobased packaging is the issue of the presented study. The conventional waste management system in the Isles of Scilly is shown in the Figure 1. Fossil-based plastic bags were assumed to be used by farmers as packaging for their produce for sale, and biowaste was assumed to be composted individually or at a municipal composting facility. In order to obtain high-quality compost, it is essential that consumers remove the packaging bags, otherwise they will end up with additional waste streams in the form of compost residues (contaminated by film particles) and plastic bags that were separated before the composting process. It is also expected that the quality of the compost will be lower and the operating costs of the system will increase.

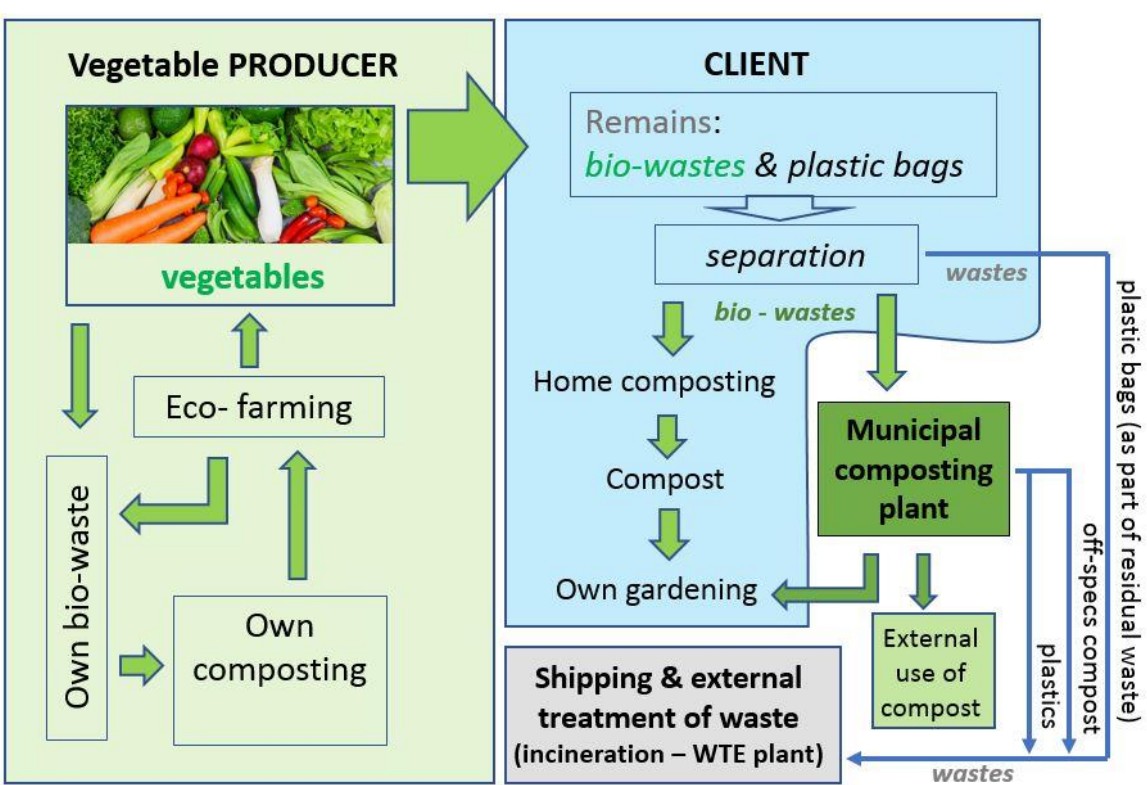

**Figure 1.** Conventional waste management on the Isles of Scilly.

When the Scilly Organics farm switches to compostable packaging, the way of handling waste also has to change. Depending on the packaging manufacturer's recommendations, biobased packaging can be disposed of by the consumer of purchased vegetables by home composting or through a local waste collection system and delivered to the municipal composting plant. The residues of vegetables in such bags could also be biologically processed. The key benefits of using biobased packaging include reduction of generated waste and its transportation, simplification of the waste management system, reduction of waste management costs, and closing the loop of green waste recycling. The sustainable management of used biobased packaging requires ecologically educated consumers properly dealing with residual waste, including compostable waste. Thus, the Council of the Isles of Scilly provides useful information supporting the inhabitants in sustainable

waste management. Special attention is paid to fossil-based plastic and biobased packaging. The Council recommends inhabitants take a careful approach to the use of alternatives to traditional plastic packaging and points out limitations and potential problems connected with the improper use of biobased packaging. Among others, the issue addresses the fact that producers' claims include the terms: "bioplastic", "biodegradable", or "compostable", which are often not supported by reliable evidence or references. Moreover, attention is paid to the situation in which the biodegradation of certain biopolymers only occurs under specific (industrial composting) conditions; thus, inhabitants cannot run the process at home successfully. The Council emphasizes that the plastic-recycling industry cannot recycle plastic alternatives deemed "biodegradable" and that these items could contaminate the recycling [48]. The Council supports the activities of the Isles of Scilly Wildlife Trust—the lead locally run conservation charity—to create a sustainable system for dealing with the elimination of unnecessary plastic packaging, re-using plastics, and changing the consumption patterns within the "Plastic-Free Scilly" priority.

## 4. Environmental Performance of the Selected Alternative Packaging for Fresh Organic Vegetables

In order to assess the main aspects of environmental performance in the life cycle perspective of the selected types of packaging for fresh vegetables (salad), an analysis of the impacts in the full life cycle was carried out, along with a literature study on the development and management of biobased packaging solutions, i.e., biobased packaging materials made from renewable resources and/or biodegradable materials.

### 4.1. The Subject of Environmental Analysis

Three types of packaging, including bags made of low-density polyethylene (LDPE), polylactic acid (PLA), and starch/polyester materials, were selected for analysis. LDPE plastic bags are currently in use by farmers from the Isles of Scilly. The company, in line with its sustainable development strategy, is planning to replace the packaging used with home compostable bags made of 100% biobased resources. In this respect, PLA is a promising biobased and biodegradable polymer that can be used as fresh organic packaging due to its good breathing properties. The starch-based packaging is an example of a material resulting from a blend of fossil-based and natural polymers. This type of packaging manufactured from 100% biologically sourced polymers is the most promising solution for farmers. That kind of material was pointed out by Scilly Organics as a potential alternative to the PE packaging used so far.

#### 4.1.1. Polyethylene Material (PE)

Low-density polyethylene is a thermoplastic material made through the polymerization process from ethylene with the use of low pressure in the production process. LDPE is rather soft, easy to shape, and transparent. After a short time of use, it becomes a waste that is extremely resistant to biodegradation, as it has a high molecular weight and contains antioxidants and stabilizers. These substances are present in all packaging plastics available on the market. They protect the polyethylene from atmospheric air oxidation in the processing stage. The presence of antioxidants makes the material difficult to decompose. Polyethylene is not biodegradable according to EN 13432 [49].

#### 4.1.2. Polylactic Acid-Based Material (PLA)

Polylactic acid (PLA) is a thermoplastic aliphatic polyester derived from renewable resources such as corn starch (in the United States), tapioca roots, chips or starch (mostly in Asia), or sugarcane (in the rest of the world). PLA possesses a wide range of desirable properties, including biocompatibility and favorable mechanical properties, and it can be molded into various shapes, making its performance comparable to petroleum-based plastics. Packaging applications of PLA include films, forms, food containers, and coatings, among many others. PLA is biodegradable and can also be recycled or incinerated. It is

used in, amongst others, the food industry to package sensitive food products. It is ideal for fresh organic packaging because it has good breathing properties. However, PLA is too fragile and is not compatible with many packaging manufacturing processes [50]. Pure PLA exhibits some limitations, such as water permeability and brittleness, and it should be strengthened with additives. Although PLA is biodegradable, PLA products should neither be littered in the environment nor composted in house composting facilities. Due to the high melting point and glass transition temperature, it requires industrial composting at 55 to 60 °C [51,52]. Due to the degradation stability of PLA products in soils at ambient temperatures, there is still a risk of environmental contamination [53,54].

### 4.1.3. Polyester-Complexed Starch Biopolymer (PCSB)

The subject of analysis is the blend of starch (34%) and fossil-based polymers (64%). This material contains fossil-based plasticizers and stabilizers. Starch is a natural polymer and is readily available from various plant sources. The main crops used for dedicated ("virgin") native starch production are maize (82%), wheat (8%), potato (5%), and cassava (tapioca) (5%) [55]. Starch is a polysaccharide that possesses an essential linear structure; nevertheless, alone it is not adequate for use in packaging because it is fragile and sensitive to environmental conditions [56]. Commercial starch materials are developed mainly for film (e.g., biodegradable packaging, bags, agricultural mulching films), injection molding, and foam applications. Biodegradable starch-based polymer is compostable in most cases in industrial or home composters, according to proper processing requirements. They can also be recycled [57]. If they are not properly disposed of, being mixed with common plastics, for example, they can be contaminated and can no longer be used [58]. Packaging made from starch blends is characterized by sufficient availability and the existence of various suppliers on the market [37].

### *4.2. LCA of Packaging Bags for Fresh Greens*
### 4.2.1. Materials and Methods

The environmental assessment of packaging was performed using the Life Cycle Assessment (LCA) method according to the ISO standard. The Life Cycle Assessment (LCA) methodology is based on the ISO 14040:2009 standard published by the International Organization for Standardization. The Life Cycle Assessment (LCA) method allows all the aspects, direct and indirect, that could potentially affect the environment to be taken into consideration and that are associated with a product or service that has been assessed. According to ISO 14040:2006, LCA is an effective tool designed for assessing the existing products, technology, and services used to identify the weakest—from an environmental point of view—points of the manufacturing process [59]. The analysis was performed using the SimaPro 8.5.2 software. In order to describe how activity datasets are linked to form product systems in the Ecoinvent version 3.4 database, the Allocation Ecoinvent default model was used. It follows the attributional approach in which burdens are attributed proportionally to specific processes.

### Goal and Scope of the Analysis

The goal of the study is to assess the key determinants of environmental performance of the selected types of packaging and provide recommendations for improving the sustainability of organic farms with regard to the selection of packaging for fresh vegetables. In the LCA performed in the study, key environmental impacts and key processes in the life cycle for the selected packaging for fresh vegetables were identified. Special attention was put on disposal scenarios analyzed according to CE rules. Selected types of bags made of biodegradable and compostable materials based on natural resources were compared with LDPE bags that are currently used on organic farms in the Isles of Scilly. The focus of the analysis is on the specific model of organic farms operating in the settings of the local community and the global market perspective and the implication of the packaging system on its final post-consumption disposal.

The assessment was done in the context of specific qualities of the biobased and biodegradable polymers. The environmental performance is also related to the functionality of the bags and the consumer behavior [60]. Obtaining good functional quality of the bags depends on the appropriate parameters, including the thickness, density, and weight of the material determining the environmental assessment.

The LCA was carried out for the case-specific scenarios developed as examples for an organic farm from the Isles of Scilly based on literature data. For these scenarios the following assumptions were made:

- The packaging is produced in European market.
- The farmer sells his organic products on a local market.
- The used packaging is disposed of according to general rules related to packaging soiled with food and is disposed of in waste bags, the content of which is intended for incineration.

In addition to the case scenarios, the reference scenarios for all types of bags were modeled based on Ecoinvent data. For the case-specific and reference scenarios the following variants were considered:

- Variant assuming the use of renewable energy resources in the PLA and PCSB biopolymer production phase.
- Variant waste-packaging treatment of in the industrial compost plant.
- Variant with energy recovery during the incineration process.

System Boundary Definition

The assessment was performed as a cradle-to-grave analysis of the different types of packaging. System boundaries clearly separate the system from the environment. The analysis included the following phases:

- Production of the raw materials;
- Production of the polymer materials from which the packaging is manufactured;
- Production of the packaging and its delivery to the organic farm;
- Final disposal of the bag and treatment of wastes.

The use phase, although very important for decision-making, was characterized; nevertheless it was not considered in further analysis because the parameters were of negligible importance due to low values. The use phase includes adequate storage of the packaging at the farm according to producers' recommendations, home delivery packing of the vegetables, and use by the consumer. The system boundaries for the three types of packaging, which served as the basis for this analysis, are illustrated in Figure 2.

Functional Unit

An LCA functional unit describes the primary function fulfilled by a product system. It enables different systems to be treated as functionally equivalent to permit direct comparison. In this study, a functional unit is defined as a packaging bag for green salad that meets the following criteria defined by the organic farm from the Isles of Scilly: transparent, holds in moisture, home compostable, made from plant materials. The packaging bag has dimensions of 25 × 37.5 cm and a thickness of 20 microns. Such parameters of the bag make it possible to contain a product (e.g., green lettuce). This is the dimension of polyethylene packaging bags currently used by the organic farm from the Isles of Scilly. The thickness is within the range of foils made from all analyzed materials, although there are opportunities to use bags of lower thickness [61].

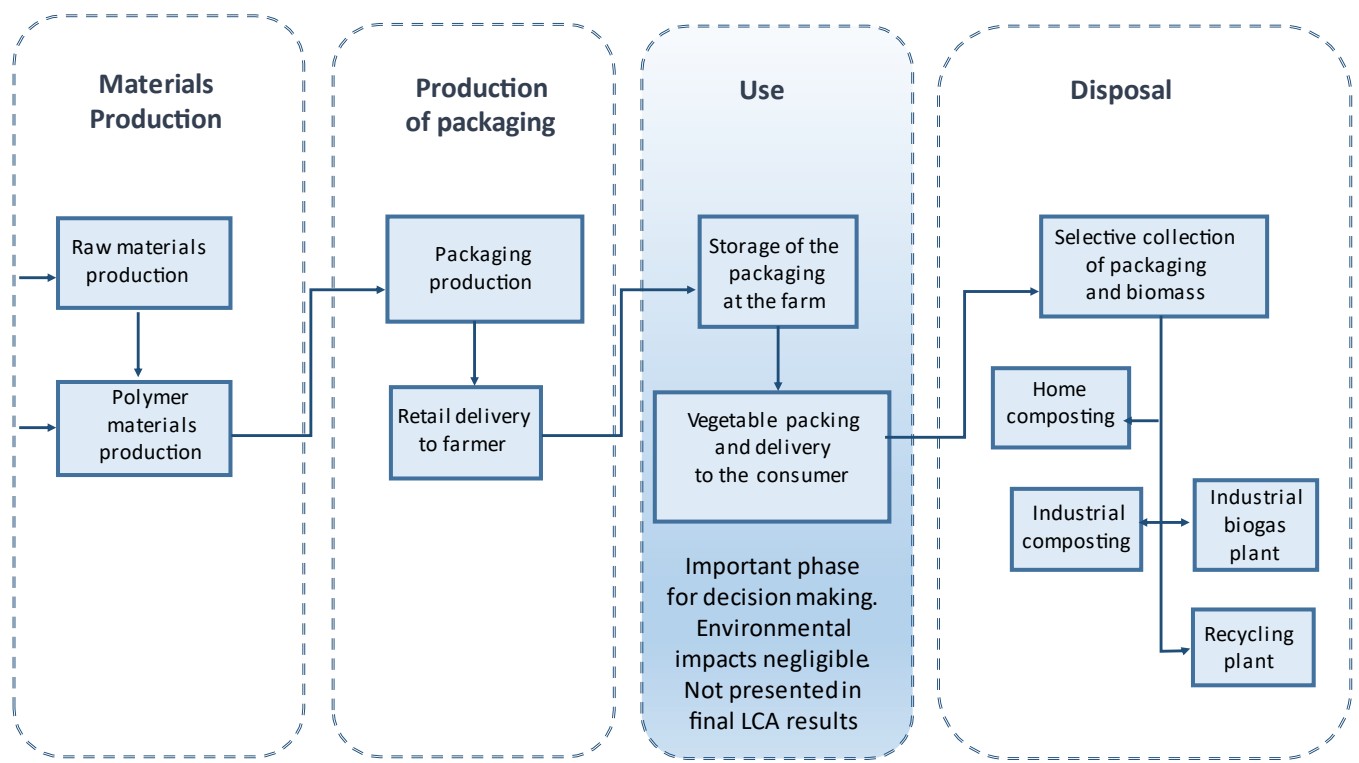

**Figure 2.** Overview of the system boundary of the LCA of the packaging.

Life Cycle Inventory

In the life cycle inventory, the bag production and post consumption utilization phases were characterized by the authors. The following factors were considered:

- Amount and type of material of which the packaging is made (raw materials used: fossil-based, biobased);
- Bag production parameters (transport, amount of energy and electricity mix in the given country of production);
- Disposal phase (transport and type of waste management);
- The inventory carried out for three types of packaging bags based on the following materials: polyethylene, low density, granulate;
- Polylactide granulate;
- Polyester-complexed starch biopolymer (PCSB) granulate.

The amount of each material was calculated based on the assumed bag geometry, volume, and material density. The production phase of bags for fresh vegetables was characterized in two ways: Firstly, the literature data were used to model specific cases representing plausible scenarios for the organic farm located in Great Britain, and the in the second approach, the Ecoinvent process for the global market was applied as a simplified reference. The reference scenarios for each type of bag were based on the foil extrusion process from the Ecoinvent database. This approach allowed the sensitivity to be evaluated and the uncertainty of the assessment to be checked.

SimaPro inventory (Ecoinvent) data were used to characterize processes in the production phase of all types of materials. Polylactic granulate production and market distribution data were used according to the Ecoinvent database and supported with additional literature data [62,63]. This process refers to the production of maize starch used as raw material, which is a product or byproduct of manufacturing processes with maize grain as feedstock. Similarly, the polyester starch biopolymer production process was applied based on Ecoinvent data. This process is also based on maize starch as a feedstock.

The three specific case scenarios for the selected types of bags were modeled based on the literature to reflect potential realistic situations for the production phase. For

this purpose, data were adapted for shopping bags for the Great Britain market made of low-density polyethylene and a starch polyester blend [64]. For polylactic acid bags, data for the case of a production site in Slovenia were applied [65]. The scenarios of the bag production refer to the following production locations: low-density polyethylene—Turkey, polylactic acid—Slovenia [66], and starch polyester blend—Norway. These include production processes, materials and energy, and transportation [67]. The specific case scenarios are described in Table 2.

**Table 2.** LCI sources for bags made from LDPE, PLA, and PCSB materials (specific case scenarios).

| Phase | LDPE | PLA | PCSB |
|---|---|---|---|
| Raw material production | Production of granulate of low-density polyethylene (LDPE) and linear LDPE according to Ecoinvent. | Global market of maize starch material according to Ecoinvent. | Global market for raw materials: Maize starch and chemicals (e.g., naphtha) include maize production and processing of maize grain based on the Ecoinvent process. |
| Polymer production | | PLA granulate production in the US based on Ecoinvent. | PCSB starch biopolymer, granulate production in Terni, Italy, according to Ecoinvent. |
| Bag production | Production in Turkey, marine and road transport of raw material from China, and road, rail, and marine transport to Scilly Island. | Bag production in Slovenia, including marine and road transport of polymer material from US and road rail and marine transport of the bags to Scilly Island. | Production in Norway, polymer material transport by road from Italy, and marine, rail, and road transport of bags to Scilly Island. |
| Post-consumer waste management | Marine and road transport of mixed communal waste to mainland and incineration. | Variant 1: marine and road transport of mixed communal waste to the mainland and incineration Variant 2: local composting at a local organic waste composting installation | |

A general description of production processes for the bags based on biopolymers was used for verification [68]. The assumed models characterize potential examples of bag production on a European scale. For the key factors determining the impacts, additional variants were considered to check the sensitivity only for the location of the bag production phase.

To characterize the disposal phase, data from the draft documents of the Scilly Local Plan 2015–2030 [46,47] were used. The waste management processes characterized in the background were based on the Ecoinvent database. The main waste output were bags with dimensions of 25 × 37.5 cm and 20 µm thickness. For the incineration process, Ecoinvent data were used, representing the European average waste-to-energy plant (WtE), which is defined based on the treatment of average European municipal solid waste (MSW). In the case of the PLA and PCSB waste management scenarios, a few approaches were applied. For PLA, the process of thermal treatment of a single waste fraction for the biodegradable waste fraction in municipal solid waste was applied as proxy, and for PCSB two proxy processes were applied: incineration of the biodegradable fraction in municipal waste (34%) and treatment of a waste plastics fraction containing PET, PMMA, and PC (66%) according to the PCSB composition (poly-$\acute{\varepsilon}$-caprolactam and starch). Additionally, PLA and PCSB Ecoinvent processes of thermal treatment were modified for PLA and PCSB based on carbon content as a reference for uncertainty analysis. For low-density polyethylene bags, the treatment of the plastics fraction containing PE, PP, PS, and PB in municipal wastes was applied.

The LCI for the analyzed bags and the specific case scenario for the disposal phase are presented in Table 3. For energy recovery in incineration, the Ecoinvent process was used. The composting process was characterized based on literature data for the

simple municipal composting process [69] in which biodegradable packaging material was assumed as organic matter and local use of compost was considered.

**Table 3.** LCI for bags made from LDPE, PLA, and PCSB materials (specific case scenarios).

| Process/Material | Unit | Bags Analyzed | | |
| --- | --- | --- | --- | --- |
| | | LDPE | PLA | PCSB |
| Raw Material and Polymer Production | | | | |
| Polymer granulate production Ecoinvent processes Own calculations | G | 3.43 | 4.538 | 4.8 |
| Bag Production | | | | |
| LDPE granulate input Data adapted from [64] | g | 3.360 | - | - |
| LLDPE granulate input Data adapted from [64] | g | 0.072 | - | - |
| Polyester-complexed starch biopolymer input Data adapted from [64] | g | - | - | 4.8 |
| Polylactide, granulate input Data adapted from [65] | g | - | 4.538 | |
| Corrugated board Data adapted from [64,65] | g | 0.253 | 0.244 | 0.259 |
| Electricity mix, consumption mix set at the point of consumption for countries of production (Turkey, Slovenia, Norway) based on Ecoinvent processes Data adapted from [64,65] | Wh | 3.217 | 3.893 | 5.118 |
| Water, deionised Ecoinvent process Data adapted from [64] | kg | - | - | 0.006 |
| Heat, central or small-scale, other than natural gas Ecoinvent process Data adapted from [64] | kJ | 4.959 | - | - |
| Transport, 3.5–16 t truck fleet average Ecoinvent process Data adapted from [64,65] and own assumptions for PLA | kg/km | 1.283 | 0.290 | 1.44 |
| Transport, 16–32 t truck, EURO4 Ecoinvent process Data adapted from [64,65] and own assumptions for PLA | kg/km | 0.708 | 9.983 | 16.8 |
| Transport, freight, transoceanic ship Ecoinvent process Own calculations based on [64,65] and own assumptions for PLA | kg/km | 17.693 | 47.975 | 7.68 |
| Transport, freight train/Europe Ecoinvent process Data adapted from [64,65] and own assumptions for PLA | kg/km | 0.991 | 1.271 | 1.34 |
| Local water transport and transport to mainland Ecoinvent process Own assumption | kg/km | 0.242 | 0.319 | 0.338 |
| Disposal | | | | |
| Municipal waste incineration—specified PE plastic fraction Ecoinvent process | g | 3.43 | - | - |
| Municipal waste incineration—biodegradable fraction Ecoinvent process | g | - | 4.538 | 4.8 |
| Cardboard recycling | g | 0.253 | 0.244 | 0.259 |
| Local water transport and transport to mainland Ecoinvent processes | kg/km | 0.242 | 0.319 | 0.338 |
| Transport, 3.5–16 t truck fleet average Ecoinvent process | kg/km | 0.221 | 0.290 | 0.308 |

It has to be noted that the assessment relies on Ecoinvent processes that are based on available specific industry information. A description of the background data related to polymer material production according to Ecoinvent documentation is presented in Table 4. According to the literature, improvements have already been made by the industry in comparison with these data. Biobased materials are extensively scrutinized in terms of their environmental performance, and the LCA results presented in the literature differ. This is caused especially by new technological and business development in the companies producing these polymers. It concerns both PLA and PCSB polymers.

**Table 4.** Basic Ecoinvent information on data for biobased polymers.

| Biopolymer | Description |
|---|---|
| Polyester-complexed starch biopolymer | Ecoinvent inventory refers to the production of granulate-modified starch. The inventory is based on calculations and extrapolations using background data from the environmental product declaration of Materbi—a range of biobased plastics produced by NOVAMONT in Terni, Italy, that are biodegradable and compostable. |
| Polylactide production, granulate | Ecoinvent inventory refers to the production of PLA. It is based on data from the world's largest PLA plant. The inventories include the LCI data from the report of the NatureWorks producer—a plant site in Nebraska. |
| Low-density polyethylene granulate | Ecoinvent inventory refers to the production of low-density polyethylene (LDPE). Data are derived from the eco-profiles of the 24 European production sites. |

Impact Assessment

For the purpose of the assessment of the environmental impacts of the packaging, the ReCiPe 2016 method was used. The ReCiPe method translates emissions and resource extractions into a limited number of environmental impact scores. These indicator scores express the relative severity on an environmental impact category. The ReCiPe 2016 method was used because it refers to both regional and global scales, presenting a wide range of impacts significant for the adopted assumptions of the analysis, in particular those related to climate change that are important for the farmer from Scilly Organics. Additionally, the Greenhouse Gas Protocol V1.01 method was used to evaluate the effect of carbon sequestration and Cumulative Energy Demand V1.09 to provide an overview of the energy used in the life cycle. The sensitivity analysis was performed considering variants in the packaging production and various waste disposal scenarios. Sensitivity assessment was performed for the case scenarios of biobased bag production for two parameters: country energy mix and transportation routes for the phases of bag production and its delivery to the consumer. The best- and worst-case scenarios of bag production for each parameter were analyzed: for transport, bag production in Western Europe (France), and for the global dimension, bag production in China, and for the energy parameter, country mix for Norway and Poland. There were no factors considered in relation to the place of polymer and raw materials production (starch and maize), but for these, general global market assumptions according to Ecoinvent were applied. To include the actual and future progress in the value chain of biobased materials in the study, additional simulations were performed using the reference SimaPro scenario, modifying the assumptions for the type of energy used.

4.2.2. Results and Discussion

Environmental Impacts

In the life cycle of the analyzed packaging, the particular impact categories differed. The most important impacts were selected for detailed analysis. The key impact categories constituting 95% and more of the total score for all types of packaging materials in the case and the reference scenarios were as follows:

- Human toxicity;
- Fossil depletion;
- Climate change human health;

- Climate change ecosystem;
- Particulate matter;
- Natural land transformation;
- Agricultural land transformation.

The environmental impact of the packaging bags (PCSB, LDPE, PLA) per 17 categories of environmental damage (midpoint analysis) in the three case scenarios is presented in Figure 3.

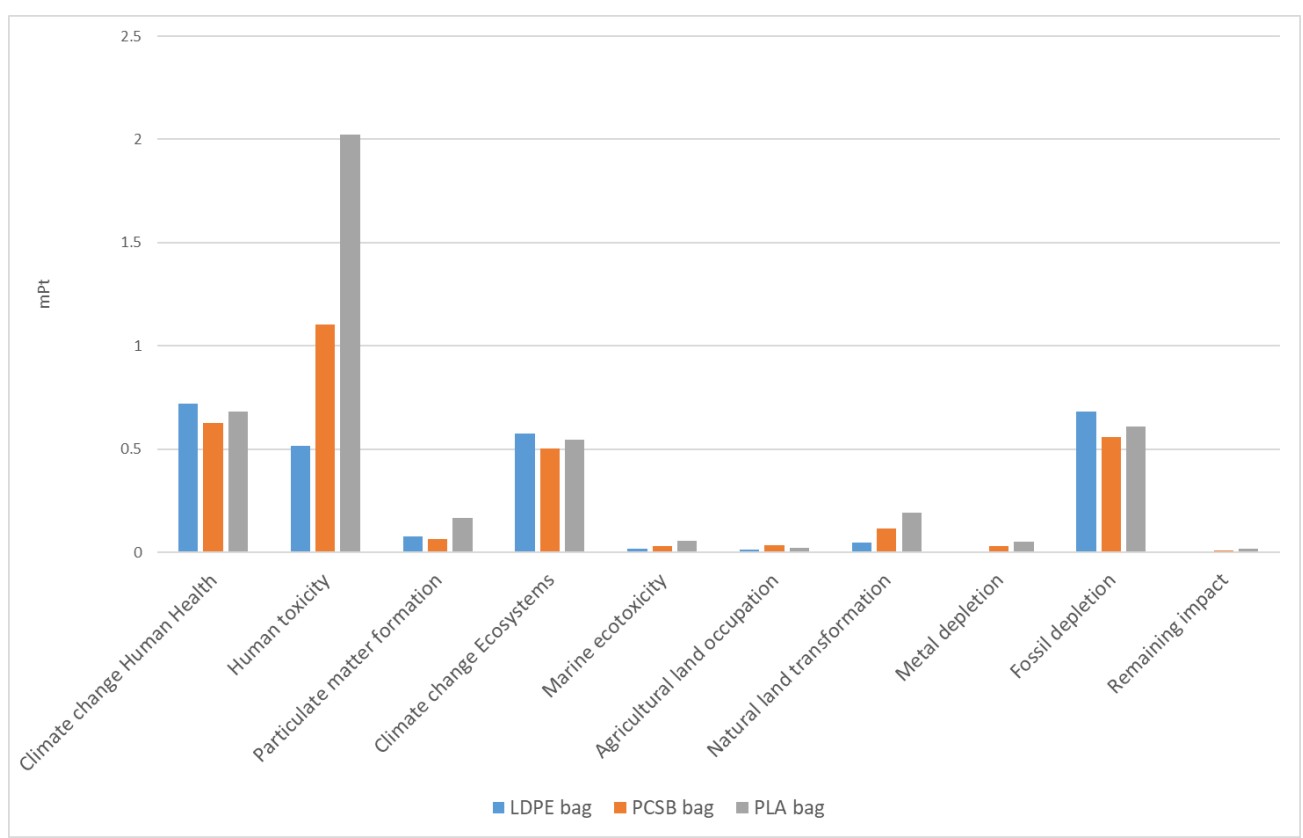

**Figure 3.** Environmental impact analysis for the packaging bags (PCSB, LDPE, PLA) per 17 categories of environmental damage (midpoint analysis) in the case scenarios.

The analysis shows that the biodegradable bags based on natural materials were characterized for the total score with higher or similar impacts than the fossil-based LDPE bags (Figure 4). The results differed in particular in the impact category. This was confirmed by the literature data [70]. It was especially pronounced for the production phase [71]. Analyzing the impacts with regard to the life cycle phases, the production of biobased polymer materials had the biggest impact in all analyzed impact categories (Figure 4). In case of PCSB and PLA, important factors were the energy used, the chemicals needed for production, and corn used as raw material. In case of PLA-based bags, the production of PLA granulate was dominant in the overall impact (weighted)—83% of the total score—and within this, the important factors were maize starch production, electricity, heat energy, natural gas, and chemical production. In the production of maize starch, the dominating factor was chemical processing and maize grain. PLA bag production constituted 16% of the total score. In the case of the production of starch polyester bags, 58% of the total impact was attributed to the production of the polymer material. On the contrary, LDPE impact in this phase constituted 41% and bag production 34%. The phase of waste disposal was of minor importance for the biobased bags. For LDPE bags it constituted 26% of the total score.

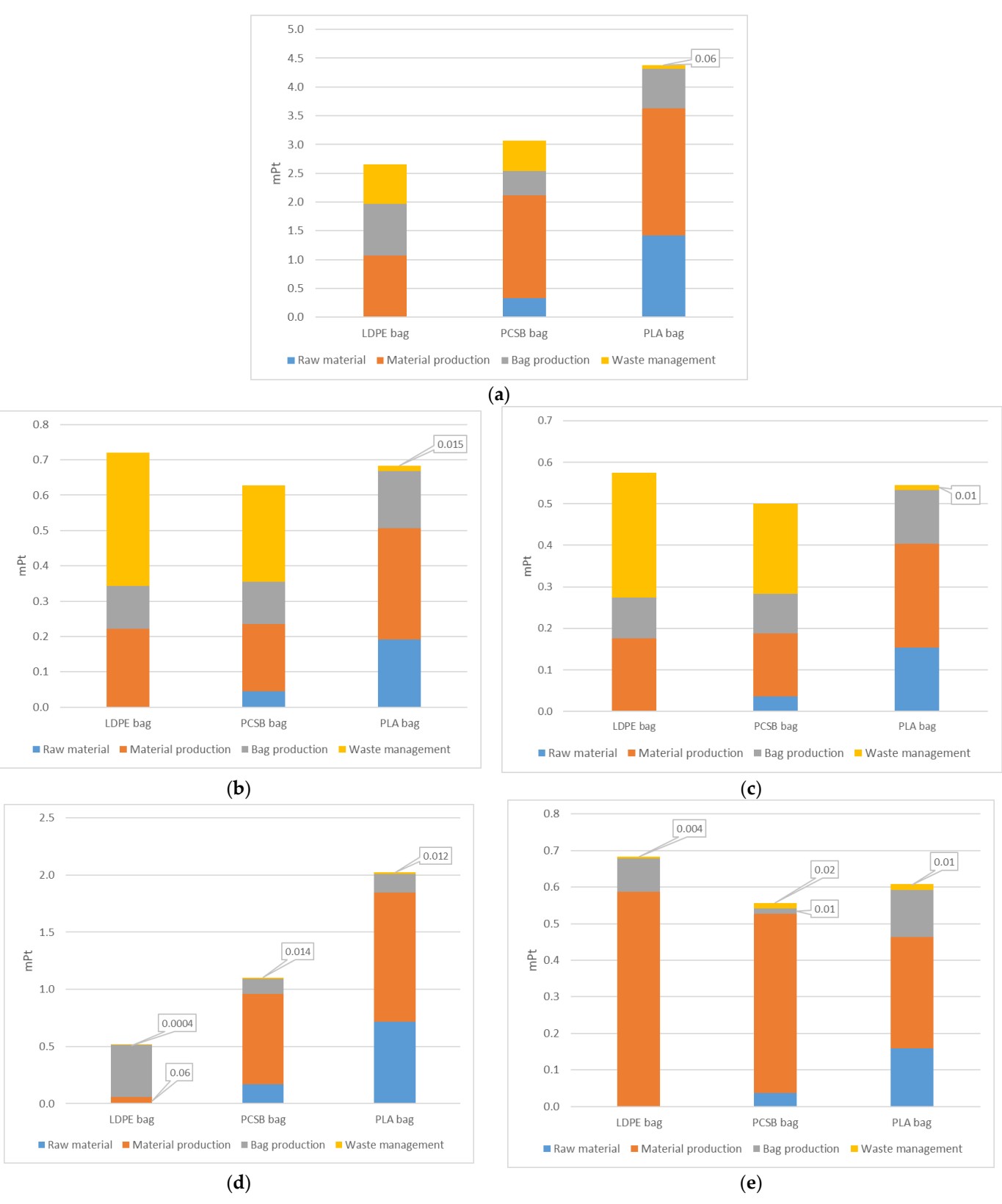

**Figure 4.** Comparison of impact characterization (case scenarios) for the selected impact categories broken down into life cycle phases (in the case of the LDPE bag the raw material and polymer production is combined). (**a**) Single score of the packaging per the LCA phases. (**b**) Climate change and human health (midpoint). (**c**) Climate change and ecosystem (midpoint). (**d**) Human toxicity (midpoint). (**e**) Fossil depletion (midpoint).

Waste management is an important factor of environmental performance in comparison with biodegradable and non-biodegradable wastes for the climate change indicator. Waste management has to be carefully analyzed in the cradle-to-grave approach [72]. LDPE bag waste management (incineration route) has a high impact expressed in the total score when the benefits of energy recovery are not included. The main factor is the incineration process. In the assessment, uncertainty related to use of the background data was considered [73]. The energy recovery in the incineration process can favor to some extent fossil-based polymers. A comparison of the impact characterization for the selected impacts broken down into life cycle phases is presented in Figure 4. The environmental impacts are expressed in mPt—a dimensionless indicator obtained after normalization and weighting expressing participation in specific impacts in the environmental load according to the ReCiPe method.

In the case of biobased bags, the most impactful phase is the production of the polymer and, to a lesser extent, the production of the bags. For PLA, raw material production is also an important phase. The phase of biopolymer production is important in all of the key impacts [63,74]. The characterization of midpoints is similar to the endpoint, with energy and chemicals playing an important role [75]. In the value chain of the packaging, the location of the particular phases of material production, packaging bag production, and the place of its final use are important environmental factors. These determine the impact due to the energy situation in a given country, and to a lesser degree, transport issues. The sensitivity related to transportation during this phase are presented below (Table 5). The table presents values of transport route simulations relative to the specific case of the studied organic farm, also including the reference, the Ecoinvent-based scenario. The influence of transportation on the assessment of the bag production phase changed in the range of 20% (LDPE bags) and 153% (PCSB bags). The high values related to PCSB reflects the high road transport burden as the main route of transportation assumed in this case scenario.

**Table 5.** Sensitivity of the LCA impact assessment for transportation schemes in relation to baseline scenario settings expressed as percentages of the change (case scenarios and the reference).

| Range | LDPE | | PCSB Bag | | PLA Bag | |
|---|---|---|---|---|---|---|
| | Total | Bag Production Phase | Total | Bag Production Phase | Total | Bag Production Phase |
| Reference scenario | −16.34 | −49.2 | 18.38 | 46.20 | −0.75 | −14.30 |
| Values higher than the case scenario | | | 5.62 | 35.54 | 21.12 | 153.32 |
| Values lower than the case scenario | −17.98 | −54.17 | −5.65 | −36.48 | | |

The value chain structure and the logistics might be more complex in reality than the analyzed settings. For the globalized scale, a high share of transportation was considered, and for the regional scale, road, rail and regional transport. The sensitivity analysis performed for various geographical dimensions justifies the global dimension of the biobased packaging market for food bags. It is a potential option for organic farms, as the maximum change in the total score of the whole life cycle was in the range of 10–13%.

Bag production processes had a generally low impact for the characterized impact category indicators. This phase is sensitive to the electricity country mix depending on the location of the production plant. Although the assessed impact of energy depends on the environmental performance of the power supply in a given country, in real conditions, site-specific conditions may also occur if a company has its own energy production system based on renewable energy [63,74]. The sensitivity of the LCA results (various country electricity mixes) markets is presented below (Table 6). The results of the simulations are expressed in relation to the baseline of the specific case scenarios, including the relation of the Ecoinvent-based reference scenario.

**Table 6.** Sensitivity of the LCA impact assessment for electricity in relation to the baseline scenario settings expressed as percentages of the relative change (case scenarios and reference scenarios).

| Dimension | LDPE Bag | | PCSB Bag | | PLA Bag | |
|---|---|---|---|---|---|---|
| | Total | Bag Production Phase | Total | Bag Production Phase | Total | Bag Production Phase |
| Reference scenario | −16.34 | −49.2 | 18.38 | 46.20 | −0.75 | −14.3 |
| Regional scale | −8.34 | −62.96 | −5.65 | −36.48 | −0.70 | −2.12 |
| Global scale | −0.37 | −2.67 | 3.72 | 13.87 | 6.39 | 19.26 |

One of the critical issues in organic farms is the greenhouse gas balance in the value chain. In the classical LCA method (as defined in ISO 14044) and the global mass balances as proposed by the IPCC, the temporary storage of carbon in biobased products is not taken into account. The main reason is that the same $CO_2$ emissions (or part of them) that are absorbed by plants are released later. In some publications/approaches, there are many proposals to introduce a discounting system for delayed $CO_2$ emissions. The widely applied specification of PAS 2050 and the ILCD Handbook specify the credit for carbon sequestration as "optional" in LCA. These optional calculations give rather different results compared to the baseline LCA method, being not fully in line with the global carbon mass balances [76].

Table 7 presents the calculation of GHG emissions according to Greenhouse Gas Protocol V1.01 for the case scenarios. The GHG emissions for the whole life cycle were the lowest in case of PLA and the highest for PCSB (case scenarios) with both factors determining the balance: production and waste management. For the biobased materials, $CO_2$ uptake was declared [77]. It has to be noted that almost all of the carbon dioxide footprint can be attributed to the manufacturing phase.

**Table 7.** Greenhouse gas emissions in the production of polymer materials per Greenhouse Gas Protocol V1.01/C02 eq. (g).

| Energy Category | LDPE Bag | PCSB Bag | PLA Bag |
|---|---|---|---|
| Fossil $CO_2$ eq | 11.02 | 19.27 | 21.40 |
| Biogenic $CO_2$ eq | 0.34 | 2.35 | 1.01 |
| $CO_2$ eq from land transformation | 0.02 | 0.01 | 0.04 |
| $CO_2$ eq. uptake | −0.33 | −4.36 | −13.71 |
| $CO_2$ eq. total | 11.06 | 17.28 | 8.74 |

The type of energy used in manufacturing essentially determines the impacts. In the case of PLA, 93% of the energy demand is attributed to PLA granulate production. To indicate the opportunities for improvement, the cumulative total energy demand was calculated. According to the Cumulative Energy Demand V1.09 method, the packaging bags made of polyester starch and LDPE were characterized with the lowest indicator values (Table 8). the CED indicator for PLA was higher. The share of renewable energy was high in the analyzed life cycle, but the performance can be further improved by using renewable energy more extensively (Figure 5) in all life cycle phases and especially during polymer production, as is argued in the literature [78]. The production of polymer composed of 100% biobased material was found to cause higher environmental burdens than fossil-based ones [79]. Changes in the production processes can also improve the overall environmental performance [61,80,81]. Moreover, waste management during the production of bags with energy-intensive materials (PLA) has to be taken into account. According to data from the Slovenian company used as an example in the study, 30% of energy was attributed to reprocessing waste PLA material. In the case of biobased polymers, the water footprint was also attributable mostly to manufacturing the polymer. In the case of PLA, it was up to 100%.

**Table 8.** Cumulative energy demand V1.09/Cumulative energy demand (kJ).

| Energy Category | LDPE Bag | PCSB Bag | PLA Bag |
|---|---|---|---|
| Non-renewable, fossil | 287.13 | 233.76 | 243.40 |
| Non-renewable, nuclear | 26.71 | 21.56 | 46.80 |
| Non-renewable, biomass | 0.014 | 0.038 | 0.083 |
| Renewable, biomass | 4.97 | 46.97 | 147.46 |
| Renewable, wind, solar, geothermal | 0.46 | 1.44 | 1.30 |
| Renewable, water | 7.41 | 28.30 | 14.79 |
| Total | 326.68 | 332.07 | 453.84 |

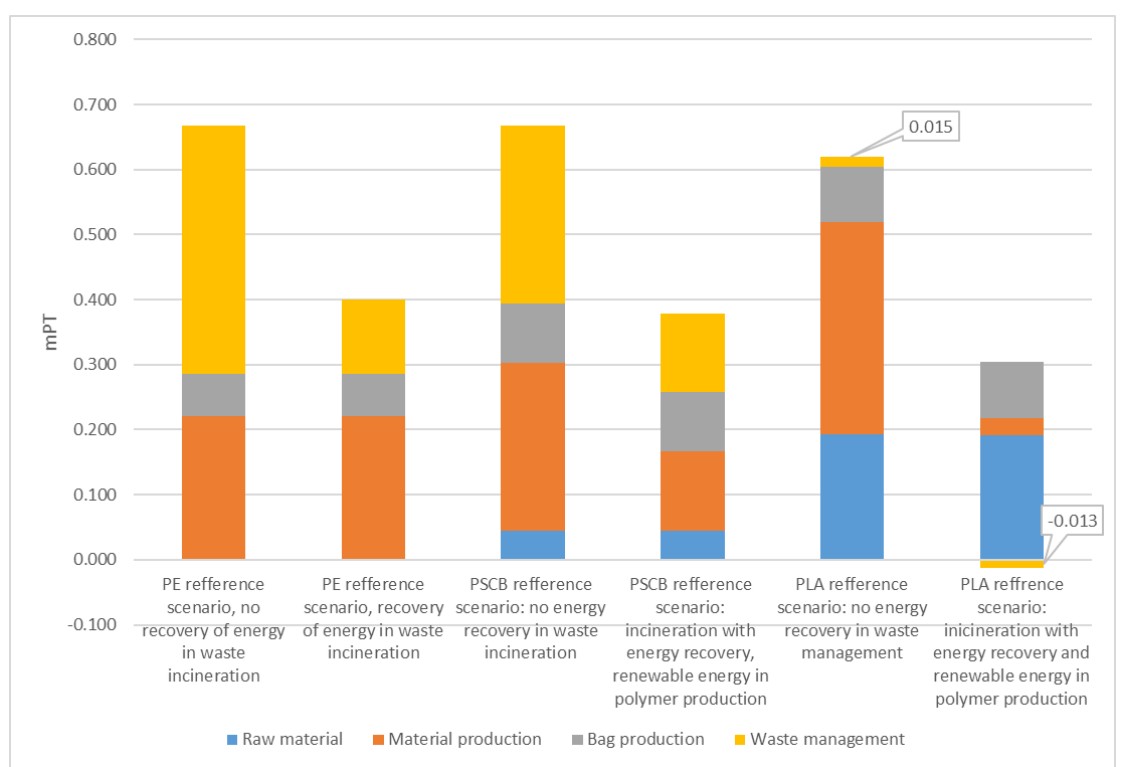

**Figure 5.** Potential of PLA and PCSB life cycle improvement in energy balance during PLA and PCSB production phase—climate change and human health (reference: SimaPro scenario of energy recovery in incineration and renewable energy use in the production phase).

Because the environmental performance depends essentially on the type of polymer material used in bag production, new opportunities should be considered by the producers. Other biobased materials such as poly-b-hydroxybutyric acid are reported as having a lower impact than fuel-based ones [82]. Other options for improving the performance of biobased bags regard the life cycles of chemicals and potential opportunities to use organic waste materials as inputs for the production of polymers and bags. One of the sources can be, for example, starch from agricultural wastes [83]. In the market survey presented in Section 3.2.3, biodegradable bags based on materials derived from non-crop plants like thistle were also found. These solutions are potentially limited, as in LCA studies of corn stover waste material and macroalgae, the authors also found that the feedstock production and biorefinery processes dominate life cycle impact profiles, with energy mix and biomass processing identified as the main influencing factors [80]. In the literature, a variety of options for improving processes in the production of polymers and foils is also indicated [84–87]. The results show that a variety of factors in the value chain

influences the environmental impacts. It makes the decision of selecting appropriate bags difficult for environmentally conscious farmers. From that point of view, improvement of the information flow on the environmental performance in the whole value chain is needed. The role of the packaging producers is important, as they are in a position to seek optimization according to sustainability rule solutions [88].

Reduction of the impacts can be achieved in the first place by lowering the volume of materials needed for bag production—especially the foil thickness and by tailoring the geometry of the bags [84–86]. The dependence of the impacts on the volume is linear. For example, the thickness specification of foils available on the market range from 10 to 30 μm. Zipper bags with additional plastic elements could also raise the impacts. Based on the market research carried out, there are offers on the market for tailored bags that can be designed and produced on demand according to client needs. This option can be a good solution for farmers, but a barrier can be posed by the relatively low amount of packaging requested for delivery and the requirements (see Section 3.2.3) for storage of biobased bags at the point of use.

According to the assessment based on the ReCiPe 2016 method, the use of organic polymer materials for packaging has a low impact on land transformation and agricultural land. It is especially small in the case of starch-derived packaging. The market represented by organic farmers oriented toward local markets is relatively small, and a rise in the impacts caused by common use of biobased packaging might not be expected in the future. The situation would differ in the case of raising the needs of biobased packaging for general consumption and other applications. A widespread switch towards organic packaging could potentially raise these impacts. The question of whether policy preferences regarding uses of biobased materials should be established arises.

In the waste management phase, the particular impacts related to LDPE are mainly attributed to the incineration process, with minor share in transport. Unlike the process of LDPE incineration, the process of biodegradable fraction incineration in municipal waste is characterized by much lower midpoint values. Energy recovery in the waste-to-energy incineration plant is an important factor (Figure 5). In the figure, it is shown that the results of the simulation of renewable energy use in the production phase and the credit of energy recovery from the waste incineration performed for PCSB and PLA lowered the environmental impact.

The differences for various modes of waste management—incineration and recycling—are important in the case of non-biodegradable materials. For biodegradable, natural-based materials, the differences between the compared treatment modes (incineration without energy recovery and composting in industrial composting installation) are not important in the cradle-to-grave analysis. The LCA results confirm the need for conscious use of biobased packaging both by farmers and consumers, taking into account the particular settings of local waste management, as in the case of Scilly Island, as presented in Section 3.2. This requires additional marketing, educational, and informational measures on behalf of the farmers and local municipalities [89–91].

It has to be underlined that the uncertainty related to use of the proxy Ecoinvent process is high. For example, for PLA, the results of waste incineration with energy recovery corrected on the base of carbon content in PLA waste in comparison to the biodegradable fraction in municipal waste showed much higher benefits, even up to 0.046 mPt for climate change, and the human health impact indicator, although for the total life cycle it was of low importance. At the same time, the high value of the total impact of this phase was approximated at the level of 0.027 mPt. The reference composting scenario for PLA bags, although on the same level as the incineration scenario in the total score, had a higher impact in the waste treatment phase. It has to be noted that the gains from home composting of clean packaging bags are limited to carbon sequestration in this process, with emissions of carbon dioxide occurring during the process. In industrial composting, additional materials, energy, and processes related to compost use have to be considered. The main gain can be achieved if the packaging bags are reused as organic matter temporary

waste bags, easing the disposal of waste by consumers. Through this, biobased packaging can play a stimulating role in promoting composting in the local community.

The reported comparison between composting and recycling of PLA showed high environmental efficiency in the latter case [92–94]. Mechanical recycling appeared to be the best solution for bio-based product disposal in order to maximize the energy savings and reduce renewable resource consumption. It is argued that recycling PLA material can improve the environmental performance of PLA packaging. However, considering post-consumer waste management systems rationality (the selective collection of bags or automatized sorting facilities in small-scale waste management systems), there is no sufficient justification in the case of organic farms acting in specific local settings that can differ according to the given farmer's business model. On the other hand, the use of home compostable packaging is desired when most of the consumers have the opportunity to compost the material. The landfilling option is viewed as being of low efficiency for PLA material with low potential for $CO_2$ sequestration [72], and is, in principle, not a recommended option according to EU waste policy. Some other techniques, including waste material processing by living organisms, can also be considered [73]. The LCA results also justify the cautious approach of the local community in promoting biobased packaging and developing local waste management installations.

It has to be underlined that for all bag types, all potential improvements, such as the use of renewable energy during the production phase and improved waste management through recycling, can be applied. From a systems perspective, there are many factors that determine environmental performance in the life cycle of the analyzed bag types. This limits the opportunities for choices of organic food producers and consumers. At this stage of biobased market development, the best situation would be life cycle realization in a regional setting with the production of biopolymers and their utilization in a dedicated, environmentally aware society. Future developments can improve the situation with market development and more investments in Europe and globally in biobased polymer production. At the same time, commitments by all countries related to the pursuit of climate neutrality by 2050 will lower the impact of energy use.

## 5. Conclusions

The selection of packaging for markets demanding high environmental quality is a challenge. This is especially challenging for organic farms, as they are frontrunners in building sustainable food production systems. Biobased and biodegradable polymers pose a promising solution for fresh organic vegetable packaging. Nevertheless, both fossil-based plastics and biobased packaging have advantages and limitations. A review of the literature as well as the LCA performed shows that there are important contexts to be considered. These are:

- The context of policy and law. Circular economy policy aims at stimulating companies to seek new packaging solutions based on natural resources in order to phasing out fossil-based ones.
- Value chain sustainability considered in the life cycle of the packages.
- Market opportunities and quality of the products in relation to specific food requirements.
- Local waste management systems, including consumer awareness and behavior.

The key aspect that has to be considered is the environmental performance in the life cycle of the packaging, taking into account all relevant factors. The conducted LCA study shows that the environmental performance of the investigated biobased packaging in general is characterized by a higher or similar impact in comparison with polyethylene bag, especially in the climate change and human risk impact categories. Despite the general conclusions, there are factors that improve the standing of biobased and biodegradable packaging. They include:

- Design of the packaging, including the amount of material used for production of the functional bag resulting from the dimensions, thickness, and density of the material type. Considering the existing market opportunities, it might be difficult to find an

optimal solution for the specific functional requirements—a certain flexibility has to be considered. Moreover, the type of material used is significant, as the impacts differ for various materials.

- Production processes and especially the amount and type of energy used for the production of the basic polymer material, which determine the most important impacts, being at the same time an essential field for improvement. In the case of PLA, most of the energy demand in the life cycle is related to PLA granulate production. Almost all of the carbon dioxide footprint can be attributed to the manufacturing phase. The amount of water used for the production of PLA polymers (the water footprint) is also attributable mostly to their manufacturing.
- Value chain characteristics, considering the place of production of raw materials, polymer material and packaging bag manufacturing, its final use, and post-consumption waste management. The location of the processes in the value chain determines the impact due to the particular energy situation in a given country and, to a lesser degree, to transport issues. The sensibility results for transport justify seeking opportunities on the global market of the packaging.
- Waste management scenario in which proper treatment of biobased waste material determines the potential for carbon sequestration and the incineration of wasted packaging. It is an important factor in comparison with fossil-based and biobased bags. Currently, the benefits in the incineration scenario favor fossil-based bags, but the changes in energy production in Europe related to climate change policy will reduce the beneficial effects in the case of fossil-based plastics.

The value chains of packaging are complex and the offer is quite diversified, which poses a challenge for farmers. There is a variety of options for improving the production of polymers and foils. The type of packaging polymers determines the quality of the waste material with regard to certain utilization pathways, such as composting either at home or in industrial installation. A producer of organic food should have the opportunity to make informed choices. For that reason, biobased packaging requires reliable information on the environmental performance of the packaging and the requirements for its proper utilization, provided by the producers and retailers. Preferably, the packaging will have certificates on the functional quality and environmental impacts. Moreover, the farmer should seek the opportunities for tailor-made solutions preventing unnecessary use of virgin materials. Environmental performance of biobased packaging could be improved by sustainable decisions made by all actors in the value chain.

The selection of the packaging bags should consider the waste treatment opportunities in the particular setting of the product consumption. Utilization of the packaging waste has to be considered in the context of the local waste management system, which can differ in particular settings. The analysis of the conditions in which organic farms operate as well as the LCA carried out shows that the use of compostable packaging meets the circular economy principles only when the local waste management system can effectively manage this kind of waste. An essential element is a selective collection system. Compostable and biodegradable packaging waste, in a locally oriented business model such as on Scilly Island, must be separated from other materials for processing and disposed of in an adequate manner. Unlike fossil-based plastic packaging, it can be contaminated with organic matter in the composting scenario. If fossil-based plastic packaging is exposed to direct contact with food and potentially contaminated with it, it is difficult to recycle and may limit its effectiveness.

Farmers have to consider both the specific requirements for waste disposal and the system of waste treatment. From that point of view, compostable bags require more engagement by the consumers and local government to secure proper disposal of the bags. Consumers properly dealing with residual waste, including compostable ones, is a crucial factor of environmental effectiveness. One issue that has to be underlined is that biodegradable/compostable bags can be a crucial stimulant for composting green leftovers from kitchen activities, and from this point of view, it has to be recommended.

**Author Contributions:** Conceptualization, B.M.-G. and J.K.; Investigation, B.M.-G., J.K., M.K. and J.S.; Writing—original draft, B.M.-G., J.K. and M.K. All authors have read and agreed to the published version of the manuscript.

**Funding:** This research is the part of CIRC4Life project "A circular economy approach for lifecycles of products and services", funded by the European Union under the HORIZON 2020 Innovation Framework Programme, project number 776503.

**Institutional Review Board Statement:** Not applicable.

**Informed Consent Statement:** Not applicable.

**Acknowledgments:** The authors are very grateful to Daizhong Su for providing support to carry out this research work.

**Conflicts of Interest:** The authors declare no conflict of interest.

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
