# Peer review of "Challenges for Sustainability in Packaging of Fresh Vegetables in Organic Farming"

_sustainability, doi:10.3390/su14095346_

Round 1

Reviewer 1 Report

Dear authors,  before accepting the manuscript please amend the following:

1) Combine all small paragraphs into larger ones, For example lines 180-197 into one larger paragraph. please check this issue elsewhere in the manuscript.

2) In your manuscript, you have so many findings with % terms, however, in the abstract none of them are mentioned, please amend the abstract and include numerical or % values to clearly indicate what were your findings. Generic findings in the abstract must be avoided.

3) Line 814 check reference!

4) Some tidying of the tables and graphs is needed, after all this is your manuscript and represents your work, so please put some effort in improving the formatting of the figures and tables as possible.

5) In Figure 5, replace the use of comma with a dot when writing numbers with decimal points. same goes for the others figures having similar issue.

6) 5. Conclusions and final remarks change to conclusions

7) It is difficult to view some of the data in Figure 3, the bars are too small to be easily viewed, it is recommended to add numbers on the top of each bar to make it clear for the readers.

8) I am a bit concerned that some of the references date back to 10-15 years ago since the study is based on 2021 values and estimations, did the authors use any data from such old references to formulate their claims and findings?

Author Response

Dear Reviewer 1,

Thank you very much your comments and suggestions. Please see the attachment with our response.

Reviewer 2 Report

The manuscript proposes a study on the life cycle assessment (LCA) of vegetable packaging in organic farming in the Isles of Scilly, UK. Different packaging materials were compared to evaluate the opportunity for improving packaging systems for fresh vegetables through biomass-based materials.

Due to the relevance of the latter issue in the context of sustainability and circular economy, this topic appears very interesting. However, the manuscript presents several critical flaws that need to be solved.

First, the title appears too long and it should be rewritten.

In the introduction, the research motivations are poorly discussed. The Authors should highlight why the current study can augment knowledge in the field of circular economy taking into account the extant studies in the literature.

Section 2 proposes a general overview of the normative framework on circularity and packaging, but scarcely considers the scientific background on these issues.

The case study is introduced in section 3, which appears too long and verbose.

Similarly, the section presenting the research methodology (materials and methods) should be clearer and more detailed, providing more information on both the tools used (e.g. the criteria used to carry out the LCA) and the case study context.

In the analysis of the results, those related to the improvement of circularity are not explained in detail. In particular, apart from the “static” LCA analysis, a wider perspective taking into account a complete life-cycle scenario should be considered (e.g. https://doi.org/10.3390/su14042235).

The discussion of results needs to be augmented, bringing to light both practical and managerial implications of the proposed study.

Finally, in the text several typos/mistakes can be found. Hence, a language review is needed.

Author Response

Dear Reviewer 2,

Thank you very much your critical comments and suggestions. Please see the attachement with our response.

Authors of the manuscript

Reviewer 3 Report

Dear authors,

I find the manuscript well written. The background is comprehensive and the practical case on an organic farm on Scilly Island is a good sustainability-study example. 

A few minor comments.

  1. In Table 3, the unit for polymer granulate, etc. should be "g" grams instead of "G", right?

  2. In Tables 5 and 6, the numbers in the first row are identical, 19.38, 46.62 ..., I believe, this must be a mistake. And, the order of plastics categories is not the same, i.e., LDPE in the last column in Table 6. Please double-check. 

  3. I suggest rearranging the order of bag categories (LDPE, PLA, PCSB) consistently throughout the Tables and Figures to improve the reader's friendliness. 

  4. I find Figure 5 not clearly explained in the article and a bit confusing.
    - The PCSB and PLA scenarios with lower impact, are they simply due to the change of renewable energy in production, and the credit of energy recovery from the waste incineration? 
    - What about the scenarios for composting? Are you suggesting that the optimal treatment of bio-related plastics is incineration with energy recovery instead of composting?

Overall, I agree this is a good LCA case study. Congrats on your efforts.

Author Response

Dear Reviewer 3,

Thank you very much for your comments and suggestions. Please see the attachment with our response.

Round 2

Reviewer 1 Report

All questions answered and paper can be accepted

Reviewer 2 Report

The Authors have satisfactorily improved the quality of the manuscript. Hence in this reviewer's opinion it can be considered for publication.

This manuscript is a resubmission of an earlier submission. The following is a list of the peer review reports and author responses from that submission.

Round 1

Reviewer 1 Report

This study conducted life cycle assessment (LCA) of vegetable packaging in organic farming in Isles of Scilly, UK. Biomass based materials were compared with fossil based LDPE. As biomass based materials are expected to play an important lore in sustainable society, this topic sounds interesting. However, due to lack of traceability, it is hard for the readers to understand the results and validity of them. Some conclusions were not supported or explained enough in the Results and Discussion.  Please see the following comments for more detail.

<Major comments>

Structure

Chapters 2 and 3 introduced background information about food packaging while chapter 4 conduct LCA of biomass based packaging in Isles of Scilly as case study. It is hard for the readers to understand the structure of this manuscript. In addition to aim of this study at lines 65-67, the authors should briefly introduce the contents and structure of this manuscript.

Chapters 1 and 2

It is good to introduce policy related to the biomass based packaging/ food packaging under circular economy in EU and UK. However, more quantitative information such as produced amount of biomass-based packaging in EU are desired.

LCA

Regarding LCA, there are many unclear points. It is good that the authors carefully considered functions provided by packaging and could set functional unit. However, the authors failed to provide process description. Although the figure 2 showed overview of system boundary, more explanations of each process needed. For instance,

l  feedstock of PLA and PS was not clearly explained. If the feedstocks were plants, the authors needs to explain cultivation process is included or not. As no inventory related to cultivation process in Table 1, it causes confusing.

l  Disposal stage in Figure 1 consisted of several treatment methods including industrial composting. However, results in 4.2.2. Result and Discussion seemed to be based on incineration. Assumptions in waste management stage needs to be explained more clearly.

l  I wonder if energy recovery was considered in incineration or not.

The readers could partly guess them when reading the results. However, clear description is essential from the viewpoint of traceability.

Line 623-632

How to calculate CO2 emissions/absorption of feedstock plants is important assumption for LCA of biomass-based plastics. Such explanations should be provided in the methodology part.

Conclusion

Some of conclusions listed here were not described in the manuscript. For instance, “According to data form Slovenian company 30% of energy is attributed to reprocessing 703 of PLA material” is firstly provided. Conclusion should be supported by the results explained before.

General

The authors should check and revise table and figure numbers through the manuscript.

<Minor comments>

Line 55

Which policy do you mean? Please clarify it.

Line 370

Abbreviation of PS generally means fossil-derived polystyrene. To avoid confusion, the different abbreviation should be used.

Line 453, Figure 2

Although the authors said that the use phase was excluded in the system boundary, it is impossible to understand it from Figure 2. Figure 2 should be revised.

Lines 499-502

In addition to the location of production phase, feedstock is also important information which affect the results. Please introduce the feedstock of PLA and starch polyester in this study.

Line 510

Last sentence seems incomplete.

Line 511 Table 1

Title of Table 1 seems strange. Contents of the table seems inventory of LCA, but some inventories such as composting are missing. Please check it and revise correctly.

Line 519

Table 1 here should be Table 2.

Line 521

Title of “Interpretation” should be renamed such as “Impact assessment” or “Environmental impacts” because interpretation is usually conducted through every LCA step.

Line 567

Figure 4 may be Figure 1.

Figures 2-6

Meaning of the unit of vertical axis in Figures 2-6 need to be explained in the manuscript.

Lines 588-590

Please revise the figure numbers in the text.

Lines 616-618

There is no evidence which support this explanation. The authors should introduce the results of sensitivity analysis to the readers.

Line 651

Please show the examples of biobased materials which have lower impact than the fuel based ones.

General

Name of biopolymers should be used consistently through the manuscript. For instance, “starch-polyester-based material (PS)” in line 369, polyester-complexed starch biopolymer in Table 1, line 519, etc. Please check them.

Author Response

Dear Reviewer 1, 

Thank you very much for your valuable comments and suggestions. We improved the text of the manuscript, trying to take into account all the critical remarks and suggestions. Please see attachment with our responses.

Kind regards

Authors of the manuscript

Reviewer 2 Report

  • Please consider reviewing the abstract. Most importantly please highlight the exact (detailed) main findings of this work, the ones in the abstract now are somewhat generic and do not provide clear idea of what was found from this work.
  • Please check the paper for English editing and typos.
  • Before the end of the introduction the authors should attempt to answer the following question: what the research gap is you found from the literature review similar or related to your work, mention it as it will strengthen the quality of your article.
  • The introduction should be one clear section, please provide more critical discussion instead of just telling us generic information from the literature, it would be better if the authors can critically compare the past literature against each other and make their conclusions about their work, also expand the introduction, it reads short and with limited information as it is now.
  • Please avoid writing small paragraphs of 4 lines or less. There is quiet few in the manuscript which can be combined into larger paragraphs to maintain good flow and readability of the article.
  • Figures 2-6 are poorly presented, they should be either renamed (a) (b) … or add each graph separately from the others, perhaps also consider better quality graphs or software to represent the data provided in them. For example, the bar for waste management is barley visible. Consider adding some numbers on the graphs to clarify this.
  • The quality of Figure 5 is poor, please replace with one with higher resolution.
  • Table 2 please check the numbers format there and the use of (,).
  • The authors are encouraged to include more detailed discussion and critically discuss the observations from this investigation with existing literature.

Author Response

Dear Reviewer 2, 

Thank you very much for your valuable comments and suggestions. We improved the text of the manuscript, trying to take into account all the critical remarks and suggestions. Please see attachment with our responses.

Kind regards

Authors of the manuscript

Reviewer 3 Report

The manuscript proposes a life cycle assessment study focused on the analysis of different packaging type for vegetables.

Actually, beside numerous mistakes and typos (e.g. language mistakes, tables and figures numbering, decimals’ notation, etc.) several critical flaws can be found.

Firstly, in the introduction research motivations are missing: i.e. it is not explained why the current study is needed and how it differs from extant studies focusing on the same topics.

The background analysis (section 2 and section 3) is not well organized: actually, it is unclear which is the practical objective of these two sections. One of the most critical issues is related to the proper differentiation from the legislative framework (for instance the European Union context) and the scientific background on circular economy and packaging ecodesign.

Then, section 4 proposes a life-cycle assessment analysis of different types of bags. This evaluation seems correct from the computational point of view, but in the literature a plethora of LCA examples can be found. Hence, it is unclear which is the novelty of the proposed study, i.e. which are the findings and practical implications.

The concluding remarks do not solve this problem and do not indicate how this study can augment knowledge in packaging ecodesign.

Author Response

Dear Reviewer 3, 

Thank you very much for your valuable comments and suggestions. We improved the text of the manuscript, trying to take into account all the critical remarks and suggestions. Please see attachment with our responses.

Kind regards

Authors of the manuscript

Round 2

Reviewer 1 Report

Although the authors revised manuscript in accordance with the reviewers’ comments, revisions are insufficient. I believe this study had enough uniqueness, however, there were many issues remained about both methodology, results, and discussions related the LCA.

Figure 2

To my last comment, the authors replied that “The use phase was characterized nevertheless it was not considered in the analysis because the parameters were of negligible importance due to the low values.” However, it is impossible for the readers to understand it from Figure 2. When the readers see Figure 2, they will misunderstand that this LCA include use phase within the system boundary.

Table 1

Although the title of Table 1 said “LCI”, it was incomplete LCI. For instance, while production phase showed electricity consumption, disposal phase provided no inventories including electricity consumption during incineration.

Lines 901-909

I cannot understand why there was no difference between incineration and composting due to lack of discussions. It sounds that we don’t need to consider waste management once the packaging was made from biodegradable materials. Such a system is not harmonized with circular economy.

Conclusions

Conclusions seemed too long. At lines 586-588, the authors said that

“The goal of the study is to assess the environmental performance of different types of packaging and provide recommendations for improving the sustainability of organic farms with regard to selection of the packaging for fresh vegetables.” When I read the conclusions, I wonder if the conclusions were supported by the results of LCA. For instance, the authors said that “Key benefits of using bio-based packaging by the Scilly Organics farm might include reduction of generated waste and its transportation, simplification of the waste management system and reduction of its costs.” It is too obvious, general information. The authors should rethink what is the motivation to conduct LCA, and what could be said from the analysis.

Author Response

Dear Reviewer 1, 

Thank you very much for your valuable comments. We improved the document according to them. Please see the attachement with our response.

Authors of the publication.

Reviewer 2 Report

The authors have answered all my questions sucessfully.

Author Response

Dear Reviewer 2, 

Thank you very much for your comments and support in working on this publication.

Authors of the publication.

Reviewer 3 Report

The Authors have improved the manuscript largely. Hence in this reviewer's opinion, it can be considered once some typos (e.g. line 949 "For example The term") are corrected.

Author Response

Dear Reviewer 3,

Thank you very much for your valuable comments. We improved the draft publication and corrected typos in it.  

Authors of the publication.